# "In an ideal world that would be a multiagency service because you need everybody's expertise." Managing hoarding disorder: A qualitative investigation of existing procedures and practices

Catherine Haighton[1]*, Roberta Caiazza[2], Nick Neave[1]

1 Hoarding Research Group, Faculty of Health and Life Sciences, Northumbria University, Newcastle upon Tyne, Tyne & Wear, United Kingdom, 2 Northumbria Healthcare NHS Foundation Trust, Newcastle upon Tyne, Tyne & Wear, United Kingdom

* katie.haighton@northumbria.ac.uk

**Data Availability Statement:** Excerpts of the transcripts relevant to the study are included within

## Abstract

Hoarding disorder is characterised by the acquisition of, and failure to discard large numbers of items regardless of their actual value, a perceived need to save the items and distress associated with discarding them, significant clutter in living spaces that render the activities associated with those spaces very difficult causing significant distress or impairment in functioning. To aid development of an intervention for hoarding disorder we aimed to identify current practice by investigating key stakeholders existing practice regarding identification, assessment and intervention associated with people with hoarding disorder. Two focus groups with a purposive sample of 17 (eight male, nine female) stakeholders representing a range of services from housing, health, and social care were audio recorded, transcribed verbatim and analysed thematically. There was a lack of consensus regarding how hoarding disorder was understood and of the number of cases of hoarding disorder however all stakeholders agreed hoarding disorder appeared to be increasing. The clutter image rating scale was most used to identify people who needed help for hoarding disorder, in addition to other assessments relevant to the stakeholder. People with hoarding disorder were commonly identified in social housing where regular access to property was required. Stakeholders reported that symptoms of hoarding disorder were often tackled by enforced cleaning, eviction, or other legal action however these approaches were extremely traumatic for the person with hoarding disorder and failed to address the root cause of the disorder. While stakeholders reported there was no established services or treatment pathways specifically for people with hoarding disorder, stakeholders were unanimous in their support for a multiagency approach. The absence of an established multiagency service that would offer an appropriate and effective pathway when working with a hoarding disorder presentation led stakeholders to work together to suggest a psychology led multiagency model for people who present with hoarding disorder. There is currently a need to examine the acceptability of such a model.

the paper, however there are ethical restrictions on sharing the de-identified data set further. The data contain potentially identifying and sensitive participant information and the authors do not have participant consent to share this dataset. Data requests may be sent to Northumbria University Research Ethics Committee (ref 1248) by contacting Laura Hutchinson [laura.hutchinson2@northumbria.ac.uk], Research Policy Manager, Research and Innovation Services, Northumbria University, UK.

**Funding:** The authors received no specific funding for this work.

**Competing interests:** The authors have declared that no competing interests exist.

# Introduction

Possessions are accumulated over time and serve an important role in defining who we are [1]. Accumulating possessions may be a learnt behaviour, ensuring survival when resources become scare [2]. However, in a minority of cases (1.5%-2% of the population) [3, 4] saving behaviour becomes excessive and disconnected from any apparent function or purpose, and the person hoards uncontrollably [2]. Hoarding disorder has been associated with excessive levels of object attachment defined as psychological or emotional bonds an individual experiences towards an object [1]. In contrast the cognitive-behavioural model conceptualises hoarding disorder as a multifaceted problem stemming from information processing deficits; problems in forming emotional attachments; behavioural avoidance; and erroneous beliefs about the nature of possessions [5].

Hoarding disorder is characterised by the acquisition of, and failure to discard, a large number of items regardless of their actual value, a perceived need to save the items and distress associated with discarding them, significant clutter in living spaces that render the activities associated with those spaces very difficult causing significant distress or impairment in functioning [6, 7]. However, some of the symptoms associated with hoarding disorder can also be part of other health problems such as physical illness, dementia, depression, alcohol and drug misuse, schizophrenia, bipolar disorder, learning disability, autism and related disorders [8].

Hoarding disorder is a social [9], economic [10] and public health problem [9] and people with hoarding disorder experience a significant reduction in quality of life [11]. Hoarding disorder increases the risk of deadly fires [12], eviction [13], pest infestation and the presence of squalor [14]. Hoarding disorder has been highlighted over the last decade as it has featured on popular television programmes and been the focus of entire television series [15] increasing public recognition of hoarding disorder [16]. Cases of hoarding disorder are addressed by multiple community services who have their own procedures and practices in relation to hoarding disorder. In supported housing for example, people with hoarding disorder create a series of challenges relating to health and safety, risk management, and safeguarding [17].

Despite limited formal evaluation [18], community partnerships [19, 20], community response models [21], community task forces [18, 22, 23] and collaborative agreements [24] are growing in popularity internationally particularly in Canada [19–21, 24, 25], the USA [18, 23, 25] and Singapore [22] as a response to the issues of hoarding disorder. These community task forces typically involve multidisciplinary teams from a diverse range of specialisations to alleviate the issues associated with hoarding disorder for individuals. However, these models have not been translated into either UK policy or practice.

In order to aid the development of a possible intervention for hoarding disorder we aimed to identify current practice by investigating key stakeholders' existing practice with regard to identification, assessment and intervention associated with people with hoarding disorder. This approach allowed us to establish the normal practice of existing intervention structure. We were then able to present a comprehensive summary of current practice to inform the development of an effective intervention. Despite emerging evidence from Canada, the USA and Singapore [18–24] it was important to develop evidence from the UK which operates a very different health and social care system. The UK national health service (NHS) is funded largely out of tax, is mostly free, comprehensive and has a provider sector that is extensively publicly owned and is more comparative to the national health systems of countries such as Italy, Spain, and Portugal.

Normalisation Process Theory (NPT) identifies, characterises, and explains key mechanisms that promote and inhibit the implementation, embedding and integration of new health techniques, technologies, and other complex interventions [26]. NPT defines implementation,

embedding, and integration as a process that occurs when participants deliberately initiate and seek to sustain a sequence of events that bring it into operation. The dynamics of implementation processes are complex, but normalisation process theory facilitates understanding by focusing attention on the mechanisms through which participants invest and contribute to them [26]. We believe that, in accordance with NPT [26], new interventions have the best chance of succeeding if they are based on an awareness and active engagement with existing organisational culture and practices therefore NPT influenced data collection methods, topic guide and data analysis. This study was developed in accordance with the Medical Research Council framework for developing complex interventions which specifies that before an intervention is piloted (as is the case for multiagency hoarding services in the UK), evidence-based modelling of the condition, its determinants and points for intervention should be specified [27].

## Materials and methods

Focus groups were carried out with a purposive sample (a non-random method of ensuring that categories of cases within a sampling universe are represented in the final sample) of key stakeholders (people with a professional interest or concern in hoarding disorder) regarding the identification, assessment and intervention associated with people with hoarding disorder. Focus groups provided socially negotiated practice examples. The study conforms to the consolidated criteria for reporting qualitative research (COREQ).

### Ethics statement

The study received approval from Northumbria University Research Ethics Committee (31/10/2017 Ref 1248) and was conducted according to the principles expressed in the Declaration of Helsinki. Informed written consent was obtained from all focus group participants.

### Sample

Key stakeholders were identified from and via an existing hoarding research group (https://www.northumbria.ac.uk/about-us/academic-departments/psychology/research/health-and-wellbeing/hoarding-research/), a multidisciplinary group (48 members) which brings together academics from English Universities, stakeholders from the Local Authorities, Housing Associations, Charities, Social Care Services, Mental Health Services, the NHS, and Emergency Services. Many of the key stakeholders were already members of this group, however members were also called upon to identify further key stakeholders (snowball sampling whereby participants suggest other individuals who could be invited to participate). Judgements about sample size, when to stop data collection and data saturation in thematic analysis are subjective, and therefore could not be determined (wholly) in advance of analysis [28] but based on previous work it was estimated that around two focus groups with 6–8 stakeholders each would provide sufficient data.

### Recruitment

Key stakeholders were invited by email by two of the authors (CH, NN) to attend a focus group to be held at a local Fire and Rescue Headquarters in North East England in November 2017. Attached to the email invite was a participant information sheet which provided further details of the research. Participation was entirely voluntary.

### Data collection

Each focus group, lasting approximately 75 minutes, was facilitated by two of the authors (CH, NN) trained and experienced in qualitive research. Focus group discussions were based on a

flexible topic guide (S1 Text) developed from the existing academic literature and discussion among members of the study team. The topic guide covered how stakeholders identified, assessed, and intervened with people with hoarding disorder. The aim of the research was discussed with the participants before ground rules were established at the start of each focus group, including confidentiality and mutual respect. Immediately following each focus group, participants were provided with a participant debrief sheet reiterating the purpose of the research. Focus groups were audio recorded and transcribed verbatim. Transcripts were fully anonymised, and recordings deleted immediately on transcription. Participants were provided with a copy of the anonymised focus group transcript to check validity.

## Analysis

Data were analysed thematically, supported by NVivo software, following the Framework method [29] with constant comparison and deviant case analysis to enhance validity. Data were repeatedly read and coded by one of the authors (CH) within a framework of a priori issues and those identified by participants or which emerged from the data. Analysis was discussed within the research team to identify areas for closer consideration and to enhance credibility of the thematic framework and interpretation.

## Results

Seventeen participants (eight males, nine females), from a range of services, volunteered to take part over two focus groups (see Table 1). No one else was present at the focus groups besides the participants and facilitators. The majority of participants were already known to the facilitators via membership of the hoarding research group and therefore had an established working relationship. Those who did not volunteer to participate (n = 31) did not provide a reason for not attending. Four themes were developed from the data. Each theme is outlined below with direct supporting quotations.

### There is no consensus understanding of hoarding disorder

Even though all key stakeholders were involved in some capacity with people with hoarding disorder there was a lack of consensus regarding how hoarding disorder was understood within their respective organisation. For many stakeholders, hoarding disorder was simply understood in terms of the risks that it posed for which the stakeholder's organisation felt they were responsible. So, for example the Fire and Rescue Service understood hoarding disorder in terms of fire risk and the complications that it could cause for both the resident and the fire service:

> "For the fire service it's quite a simple one. It's anything that is going to cause us problems in the event of a fire. So, although we are concerned with the person, their health and wellbeing, for us it's predominantly the fact that the hoarding causes an issue in terms of fire risk, which potentially will trap the person in their property, and it will also cause issues for the firefighting activities that go on if they do have an incident." ID10 Fire and Rescue Service FG2

Many stakeholders reported that hoarding disorder was only identified as a problem if it posed a serious risk to their assets, to their organisation, to others or to the individual with hoarding disorder:

> "Sometimes we do find cases where there is hoarding, but as long as we deem it to be not a risk, not impacting on the property, themselves in that sense, the neighbours, we let it go." ID15 Council FG2

**Table 1. Characteristics of focus group participants.**

| Focus Group | ID | Gender | Role | Description | Organisation Type |
|---|---|---|---|---|---|
| 01 | 01 | Male | Mental Health Recovery Support Community Outreach Team | Supports council tenants with mental health problems in the community to live as independently as possible | Council |
| 01 | 02 | Female | Clinical Psychologist | Supports people, aged 65 or older, seeking specialist assessment and treatment for a mental health difficulty | NHS Trust |
| 01 | 03 | Female | Trainee Clinical Psychologist | In training to work with people of all ages on a wide range of psychological difficulties in mental and physical health | NHS Trust/ University |
| 01 | 04 | Female | Wellbeing Coach | Works with people, aged 50 or older, who hoard to dangerous levels | Charity |
| 01 | 05 | Male | Housing Solutions | Coordinates housing register | Council |
| 01 | 06 | Male | Policy and Strategy Manager: Prevention | Home safety checks | Fire and Rescue Service |
| 01 | 07 | Female | Social Worker | Long standing interest in mental health, mental capacity, and adult safeguarding | University |
| 01 | 08 | Male | Clinical Psychologist | Works with children and adults with a variety of mental health difficulties | NHS Trust/ University |
| 01 | 09 | Female | Safeguarding Adults Consultant/ Trainer | chairs/authors safeguarding adults reviews and safeguarding adults board | Consultancy |
| 02 | 10 | Male | Community Safety Manager | Responsible for reducing accidental fires and fire deaths through a wide range of prevention activity | Fire and Rescue Service |
| 02 | 11 | Female | Mental Health Social Worker Team Manager | Provides services to individuals who experience mental illness and may have complex, multiple needs | Council |
| 02 | 12 | Female | Older Persons Services Manager | Manages independent living schemes and emergency response service | Housing Association |
| 02 | 13 | Male | Housing Manager | Responsible for antisocial behaviour, safeguarding and hoarding disorder | Housing Association |
| 02 | 14 | Male | Clinical Psychologist | Works with people of all ages on a wide range of psychological difficulties in mental and physical health | NHS Trust/ University |
| 02 | 15 | Female | Neighbourhood Housing Operations Manager | Responsible for housing services | Council |
| 02 | 16 | Female | Head Of Housing Support | Responsible for antisocial behaviour, safeguarding and supported housing | Housing Association |
| 02 | 17 | Male | Solicitor | Adult social care law, mental capacity, court of protection, deprivation of liberty law, mental health law. | Council |

Risks, to some of these organisations, from people with hoarding disorder, were understood in terms of their legal responsibility for preventing the possible negative outcomes from the disorder:

*"I think for us as a landlord we are also concerned about the individual but also to do with risk. Corporately speaking from a social landlord who had a tenant killed. . .we can't afford that to happen again. And that risk trickles through to corporate organisations like us, to the fire service and if it goes wrong we are all liable."* ID13 Housing Association FG2

Stakeholders also reported that they were seeing different types of hoarding, in particular 'ordered' versus 'chaotic' hoarding, with the former posing less of a risk and therefore being less likely to be identified as a problem:

*"If you can get to the exits that means we can get in they can get out. We can see the state of the electrics. All the bits that would cause a hazard are ok. Ok actually they've got a load of stuff but what's the chance of then having a fire. Somebody who's got stuff strewn all over we can't see what's plugged in, we can't see where the fires are and they can't get to the exits."* ID10 Fire and Rescue Services FG2

Within both focus groups there was also considerable discussion relating to ordered hoarding and collecting. When one participant understood hoarding disorder as *"when a collection has taken over the home"* (ID16 Housing Association FG2) it sparked debate on the importance of making a clear distinction between the two concepts as this could impact on availability of support:

*"Well, it is very difficult, because in research they have defined the difference between a collection and a hoard, and there is very distinct difference that are important, because if someone is collecting, they are not eligible for safeguarding or services."* ID09 Consultancy FG1

This was confirmed by a stakeholder who reported how a case of ordered hoarding was not considered to be a problem by a judge when their housing association took their tenant to court:

*"Now everything is ordered and we've taken her to court and lost and spent a lot of money in losing that case. The district judge said it's her article 8 human right to live like that if she chooses to do so."* ID13 Housing Association FG2

Many stakeholders reported hoarding to be to be either a "symptom of" (ID10 Fire and Rescue Service FG2; ID17 Council FG2) or a "solution to" (ID11 Council FG2) some other problem making it difficult to treat unless the underlying issue was correctly identified. Many of the health professional stakeholders in the sample reported the importance of correctly diagnosing hoarding disorder so that they could appropriately intervene. It was reported that hoarding could be a symptom of issues with memory (such as dementia), frontal lobe damage, problems with executive functioning, autism, a range of other mental health problems, social issues (not linked to trauma), motivational issues, or trauma:

*"When someone says we have a referral for hoarding, the first thing I think is, is it actually hoarding, is it some kind of memory issue, because I work with older people, or is it something more the social issue that is not linked to trauma."* ID02 NHS Trust FG1

Stakeholders reported that there was considerable stigma associated with people with hoarding disorder and stressed the importance of viewing people with hoarding disorder in a non-disapproving manner to improve intervention:

*"And I think part of the difficulty. . .is accepting that we've all got a particular view, usually a pejorative view, as to someone who hoards and until we can actually address that in ourselves and get over that, we are always going to approach that person or that situation in a particular way."* ID11 Council FG2

### Cases of hoarding disorder are on the increase

Stakeholders reported a wide range of estimates of the number of current cases of hoarding disorder within the region from 24 to 2000. However, all stakeholders agreed that number of cases appeared to be increasing:

*"What came from your team about the people that have been identified with possible hoarding, in terms of fire risk, it was more than 2000. . .and from us mental health, first year I started, I only had the one referral, but now this has been going on, I think we have had 6 referrals just this week for people with hoarding."* ID02 NHS Trust FG1

Increased identification of cases was reported to be because of publicity about the disorder and TV documentaries about hoarding disorder in addition to changes in targeting of services:

*"I think, more worryingly, is that because of our change of targeting, because we are quite often a service that is first through the door because there is a type of home safety check, free service that we offer. Because we are. . .focused on the older person, social isolation, highly vulnerable, known possible live alone. . .what we have found, definitely an increase in the numbers of hoarders and issues with cluttering."* ID06 Fire and Rescue Service FG1

There was some agreement that the number of cases of hoarding disorder might appear higher in social housing because it was more likely to be identified there:

*"It's because he is in social housing, and every year someone is going in there for one reason or another. Either to do an annual gas safety check or an electrical check because we are duty bound to do that. . .That is probably why we have more than we know of in social housing than we do in other sectors, because we just don't know, it's not that they are not there."* ID16 Housing Association FG2

## People with hoarding disorder are more likely to be identified in social housing

The Clutter Image Rating Scale was most commonly used to identify people who needed help for a hoarding disorder, in addition to other assessments relevant to the stakeholder:

*". . .the clutter scale and then we try and screen the memory using a cognitive assessment, we try do some anxiety, depression assessment, to make sure that we have got everything, to identify, so we try and have a battery of multi, different."* ID02 NHS Trust FG1

*"Purely the clutter scale and the perception of fire risk."* ID06 Fire and Rescue Service FG1

Stakeholders reported the importance of assessing capacity from the outset followed by a best interest assessment:

*"You know capacity, we have this particular test, that we have to apply. If you are within the test, you know and lack capacity, in legal terms, then we can do things for them. If you are outside, then the law says we can't. . .So a best interest's assessment follows a capacity, if you have capacity, then you know, the question is what do you want to do, as the person that gets to say what happens. . .you may well be doing exactly the same for someone who has capacity and tells you what they want. And someone who lacks capacity you decide on their behalf. But it might be that they are entirely at odds, and it might be that we can't work it out as professionals and the person or family might think differently, and we might have to ask a court"* ID17 Council FG2

As reported earlier, people with hoarding disorder were commonly identified in social housing where access to the property was required for annual safety checks, secure tenancies (a tenancy which does not have a fixed term or end date) were common and it was difficult to remove tenants without safeguards and scrutiny. However, identifying hoarding disorder in social housing presented a problem to staff when differentiating between hoarding disorder and general neglect of property:

*"And one of the challenges that we've had is trying to differentiate for housing office staff the difference between what we would class as a "condition of property" type of case and a hoarding case because they've got massive differences and some of our officers are going into homes and seeing some home that not well kept and might be a bit cluttered and untidy as not a hoarder and what the difference is."* ID16 Housing Association FG2

Stakeholders reported that people had less time and opportunity to hoard in privately rented housing and that people living in their own property with hoarding disorder were only likely to be identified if a neighbour complained and an environmental health officer, adult social care, the fire, police or ambulance service, or the Royal Society for the Prevention of Cruelty to Animals (RSPCA) became involved and even then, this might not provide a long-term solution:

*"Again though, that would be something that is very black and white, stop doing that to the animals or tidy up your backyard, right ok, we are gone again. Until you cause someone else a problem you are not a problem, whereas, with social housing and landlord properties, it is, it is kind of an ongoing thing because we will come back next year."* ID10 Fire and Rescue Service FG2

One stakeholder raised the importance of referring in the correct way so that people with hoarding disorder could be appropriately identified:

*"There is a problem in the way that safeguarding referrals come in from people like fire, police and ambulance services because, there is a difference between a safeguarding referral and a vulnerable adult concern form, so we could get in from a local authority, 2000 vulnerable adult concern forms in a month, whereas, we could get maybe, 40–50 safeguarding referrals, but amongst those vulnerable adult concern forms there might be self-neglect that they are not identifying as safeguarding. And they just kind of may get missed or lost or, whatever."* ID09 Consultancy FG1

## There was support for a multi-agency approach to intervention

Stakeholders reported that the symptoms of hoarding disorder were often tackled by enforced cleaning, eviction, or other legal action:

*"So, everybody would like to see that long-term solution but the reality is, I can't wait for long-term solutions, so I am going to get together with [housing association] and we will push for an eviction, we will have some support services there if we need it. . .but in the meantime I am going to work with [housing association] to get an eviction or to force a clear up."* ID10 Fire and Rescue Services FG2

However, stakeholders reported that these approaches were extremely traumatic for the person with hoarding disorder:

*"We have a gentleman that we are working with at the moment, and he has been constantly threatened to be evicted because his house is in a terrible state. . .we have tried to explain. . .that actually it would be very traumatic for him to go in and just get rid of things."* ID02 NHS Trust FG1

And failed to address the root cause of the disorder:

*"But we do that, we do that, we go in, we declutter, we clear out, we get an order, but if you don't address the underlying cause, it just comes back again. We have all had the cases don't we, where a couple of years later, it is like oh here we go again."* ID16 Housing Association FG2

Ultimately this resulted in the person with hoarding disorder simply being passed around services:

*"I manage now our homeless service in [location] as well that is under our organisation now which is relatively new, but our team that work in the council with private landlords, had took some proactive action around some hoarders in the private rented sector, and ended up serving legal notice and those families were subsequently evicted with no discussion with ourselves prior to that, and so we ended up then with several families coming to us all at once, to say I am homeless, I have been evicted because of this situation. When we looked into it, it was hoarding. We then had a duty to rehouse those families because of all their vulnerabilities, so we have now got those hoarders. The hoarding has never been addressed; we have just inherited the problem."* ID16 Housing Association FG2

Many stakeholders reported that once they had identified a person with hoarding disorder there was a lack of engagement or support from other services:

*"And I think that from the housing perspective what we've sometimes struggled with is to, is to get that involvement from some of those other specialist support agencies that can start unpicking that. A housing officer can't go in necessarily and unpick that with an individual and start having that conversation necessarily because more harm could be done than good in that."* ID16 Housing Association FG2

Stakeholders from councils and housing associations reported that social workers and health professionals such as psychologists or psychiatrists would rarely engage with people they identified as having hoarding disorder as they were unable to make referrals directly to mental health services. Stakeholders reported that even when other services were involved *"...there's no real working together".* ID12 Housing Association FG2. While stakeholders reported there was no established services or treatment pathways specifically for people with hoarding disorder, stakeholders were unanimous in their support for a multi-agency approach to working with people with hoarding disorder believing this would be the most cost-effective approach:

*"That's why it is fundamental to always have a multi-agency approach and...if you see hoarding, from my point of view, you should start getting the team ready to go, and then see who is needed...so from our team we always go, as I said, psychology, CPN and support worker because you have got the medical side of things, so if we need to get the psychiatrist involved the CPN can do the initial assessment already, so we speed that up, the support worker can help the clearing because they can help the initial process, and I can work around the trauma. And then if we need to get social services, then they are already aware that we are going to go in, so it is always about having that plan ready to start with because, you can't be not prepared for what is going on, because these people have been let down, for years. So if you go in and be another one that lets them down, then you are just going to, you have lost them in the beginning."* ID02 NHS Trust FG1

Many stakeholders reported misunderstandings from both the public and within their own organisations particularly regarding timeframes for dealing with people with hoarding disorder:

*"I think there is a certain school of thought out there that thinks that we can just, you know, send them a letter and if they don't comply we can just go in and look them out so to speak which obviously we can't do. . .They expect us [to just turn up] and you know, obviously it can take two to three years to work with a particular individual but at the time the neighbours think well the council don't do anything and they are not interested, they are not bothered."*
ID15 Council FG2

## Discussion

In our focus groups with key stakeholders from housing, health, emergency services, and social care there was no clear consensus for what constituted a hoarding disorder presentation and therefore each service followed different procedures to manage clutter. Organisations were focused on quickly resolving hoarding disorder by decluttering properties, eviction and other legal action. The repetitive clearing cycles imposed by authorities were traumatic and deleterious, whilst also being ineffective and costly for the various services involved. Organisations focussed solely on the issue most pertinent to them, and this is a major barrier to addressing the underlying causes and multifaceted nature of hoarding disorder. A major concern was the lack of multiagency working, which led to complications managing risk for the individual, the wider public and services dealing with self-neglect. There was significant support for a multiagency approach to hoarding disorder.

One of the earliest reported multiagency approaches, the Clark County Hoarding Task Force (CCHTF) USA, was formed to address the health and safety issues associated with hoarding disorder [23]. The CCHTF was composed of numerous community agencies and groups, including the Health District, local animal cruelty and rescue resources, Code Enforcement, Sheriff's Office nuisance officers, Adult Protective Services, United Senior Services, the prosecutor's office, and Mental Health Services, all willing to work together [23].

Research has suggested that interventions conducted by multiple agencies can be especially valuable [30]. Multiagency approaches have been found to help with hoarding disorder and the associated economic and social costs, for example, an integrated community response model for the delivery of resources and support was developed in Edmonton, Canada to provide sustainable support and services to people with hoarding disorder [21]. Authors reported that working together collectively, in a multi-disciplinary fashion, allowed individuals living with hoarding disorder to be respected and provided support to ensure successful intervention [21]. Authors proposed that addressing hoarding disorder through an integrated approach across the lifespan would reduce burden on the health care system [21].

Bratiotis [18] carried out a qualitative study assessing five hoarding task forces in the USA, drawing on perspectives from mental health, housing, social services, emergency services and health agencies. While the task forces showed promise in reducing the negative outcomes of hoarding disorder in the short-term, they were often formed on an ad-hoc or case-by-case basis and lacked a single organisational control mechanism and full organisational commitment. In contrast our study showed significant support for multiagency working and consensus among agencies that this should be a psychology led approach.

While stakeholders in our study did not report issues with children being affected by hoarding disorder Bratiotis [31] noted that multidisciplinary coordination of interventions should

include professionals who support children and family as well as the person with hoarding disorder when working with a family with children affected by hoarding disorder,

A hoarding task force and relevant legislation was also introduced in Singapore to address the issue of hoarding disorder in the community [22]. The task force involved the Ministry of National Development, Ministry of Health, Ministry of Social and Family Development, police, Housing and Development Board, Singapore Civil Defence Force, National Environment Agency, People's Association, and Institute of Mental Health [22]. The task force brought together the expertise and power to better tackle the issue of hoarding disorder in the community [22]. However an exploratory qualitative study conducted among the hoarding task force service providers revealed that they possessed limited authority and enforcement power [32] and authors noted that in the absence of therapeutic or medical intervention and/ or community support to address the root cause, hoarding disorder may recur in the cases referred to the task force [22]. This data from Singapore further supports our finding that a multiagency model should be psychology led therefore allowing the approach to address the hoarding disorder to be therapeutically driven.

A collaborative agreement between the main stakeholders (municipal, fire, police, public health and the regional health centre) providing services for the management of severe cases of domestic squalor in rural and semi-urban areas located in Quebec, Canada provided a more personalised and coordinated case management which took into account the individuals environment [24]. This approach decreased the risks associated with a cluttered dwelling as well as allowing the stakeholders to cope with varying degrees of health risks associated with various medical problems [24].

More recently Bodryzlova [20] described a community partnership created to improve the clinical management of hoarding disorder in Canada. The Montreal Compulsive Hoarding Enlarged Committee (MCHEC) brought together 30 partners: service users, health and social care professionals from public and non-profit sectors, and municipal service providers and researchers which contributed to the creation of a common language, attitudes, and expectations among the professionals dealing with hoarding disorder; authors concluded that MCHEC provided the best clinical practices for treatment and advocates for the dignity of the people affected by hoarding disorder [20]. Another approach, the Hoarding Action Response Team (HART) model of a community-based intervention for hoarding disorder involved a partnership between fire prevention and public health in Vancouver, Canada [19]. HART was associated with a reduction in clutter and preservation of tenancy and authors concluded that effective solutions could be realised with a cross-disciplinary and committed team that places the client at the centre of the intervention [19].

Our findings confirm those reported in other similar research for example in a survey of 236 social service staff members in Florida, USA about their experiences with cases involving hoarding disorder respondents reported that hoarding situations were difficult to resolve and involved multiple community agencies [33]. Authors suggested that collaborative interagency protocols were needed to manage hoarding disorder [33]. Another survey of primary mental health services in Quebec, Canada reported health and social services professionals lacked hoarding disorder clinical management tools, training and formal collaboration with municipal (housing, building security, fire prevention) specialists [34]. Once again authors suggested the need for formal collaboration with municipalities and community organisations in order to improve services for people with hoarding disorder [34].

As far as we are aware this is the first UK study to examine stakeholder's procedures and practices for managing people with hoarding disorder to be able to suggest a psychology led multiagency model for future management, support and treatment. This study not only adds to the existing body of evidence for multiagency approaches to managing hoarding disorder in

Canada, the USA, and Singapore but, as the UK operates a very different health and social care system, provides more generalisable evidence for countries with comparative health systems such as Italy, Spain, Portugal, Malta and New Zealand. However, we must also acknowledge the limitations of our approach. While our sample of stakeholders was relatively small, there was a wide variety of organisations involved; however there were some services which were not represented such as the ambulance service and the police. While not every key stakeholder was represented the fact that 17 stakeholders gave up their time to attend these focus groups represents their interest in and commitment to support people with hoarding disorder.

## Conclusion

The absence of an established multiagency service that would offer an appropriate and effective pathway when working with a hoarding disorder presentation led stakeholders to work together to suggest a psychology led multiagency model for people who present with hoarding disorder [35]. Caiazza et al. [35] suggested that hoarding disorder requires input from multiple sources, including emergency services, social services, and mental health services. While this multiagency model has only been implemented on an *ad hoc* basis this has led to positive outcomes backed by case studies [36, 37]. The Ottawa Community Response to Hoarding Coalition similarly recommended that one agency be selected as the central coordinating unit in the network of support for people with hoarding disorder [37]. However social workers or community nurses are typically the chosen coordinators of care, navigators of the system and advocates for patients [38, 39]. Here we recommend a psychology led multiagency model that would allow for diagnosis of hoarding disorder as well as coordination of care based on each patient's mental stability to cope with intervention. There is currently a need to examine the acceptability of the psychology led multiagency model.

## Supporting information

**S1 Text. Focus groups topic guide.**
(DOCX)

## Acknowledgments

In addition to all focus group participants, we would like to thank Mia Campbell, George Murray, Vincent Deary, Carole Southall, and Markuu Wood who helped us facilitate the focus groups.

## Author Contributions

**Conceptualization:** Catherine Haighton, Roberta Caiazza, Nick Neave.

**Data curation:** Catherine Haighton, Roberta Caiazza, Nick Neave.

**Formal analysis:** Catherine Haighton, Roberta Caiazza, Nick Neave.

**Investigation:** Catherine Haighton, Roberta Caiazza, Nick Neave.

**Methodology:** Catherine Haighton, Roberta Caiazza, Nick Neave.

**Project administration:** Catherine Haighton, Roberta Caiazza, Nick Neave.

**Validation:** Catherine Haighton, Roberta Caiazza, Nick Neave.

**Writing – original draft:** Catherine Haighton, Roberta Caiazza, Nick Neave.

**Writing – review & editing:** Catherine Haighton, Roberta Caiazza, Nick Neave.

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
