## [Decision Letter · Decision Letter 0]

7 Sep 2022

PONE-D-22-14253“In an ideal world that would be a multiagency service because you need everybody’s expertise.” Best practice for managing hoarding disorder: A qualitative investigation of existing procedures and practices.PLOS ONE

Dear Dr. Haighton,

Thank you for submitting your manuscript to PLOS ONE. After careful consideration, we feel that it has merit but does not fully meet PLOS ONE’s publication criteria as it currently stands. Therefore, we invite you to submit a revised version of the manuscript that addresses the points raised during the review process.

Please note that we have only been able to secure a single reviewer to assess your manuscript. We are issuing a decision on your manuscript at this point to prevent further delays in the evaluation of your manuscript. Please be aware that the editor who handles your revised manuscript might find it necessary to invite additional reviewers to assess this work once the revised manuscript is submitted. However, we will aim to proceed on the basis of this single review if possible. The reviewer has made a number of recommendations to improve the manuscript, including discussion of a broader range of international literature on this subject. Please respond carefully to this and all the points the reviewer has raised when preparing your revision.

We look forward to receiving your revised manuscript.

Kind regards,

Jamie Males

Editorial Office

PLOS ONE

Journal Requirements:

Reviewers' comments:

Reviewer's Responses to Questions

**Comments to the Author**

1. Is the manuscript technically sound, and do the data support the conclusions?

Reviewer #1: Partly

2. Has the statistical analysis been performed appropriately and rigorously? 

Reviewer #1: N/A

3. Have the authors made all data underlying the findings in their manuscript fully available?

Reviewer #1: Yes

4. Is the manuscript presented in an intelligible fashion and written in standard English?

Reviewer #1: Yes

5. Review Comments to the Author

Reviewer #1: “In an ideal world that would be a multiagency service because you need everybody’s expertise.” Best practice for managing hoarding disorder: A qualitative investigation of existing procedures and practices.

The study presents the attitudes and practices of key intervenors in case management of hoarding behaviour. Authors point out that the management of hoarding behaviour in the community is difficult, that attitudes and practices of stakeholders are homogeneous, and that creation of the multiprofessional team might be difficult.

It is a timely and needed study. Its only limit is that it omits the wide international experience in creating multi-professional teams for hoarding disorder around the world. An extensive body of academic and gray literature exists on the topic. Integrating this literature into the introduction and discussion would greatly improve the value of the current work.

In addition, the reading of the paper by a clinical psychologist is desirable in those which concern the distinction between hoarding behaviour and hoarding disorder, as well as more ancient and more recent cognitive models of hoarding disorder.

Title and abstract.

“In an ideal world that would be a multiagency service because you need everybody’s expertise.” Best practice for managing hoarding disorder: A qualitative investigation of existing procedures and practices. Short Title: Best practice for managing hoarding disorder – The title is incorrect. The term “best” practice is the

I would suggest avoiding the term “best practice” as misleading: it evokes clinical guidelines and systematic reviews.

Introduction.

LL 35-36. “’ Possessions are accumulated over time and many of these items are given sentimental value”.

See reasons for hoarding in Moulding, Knight, O'Connor. Also, see the cognitive model of hoarding in Frost and Hartl.

LL 36-37. “such behaviour may be adaptive by ensuring survival when resources become scare”

Interesting tackle, but still unapproved. Sources are needed

LL 38-39 However, in a minority of cases the normal hoarding tendency becomes pathological, and the person hoards uncontrollably

See Nordsletten for the exact proportion of problematic hoarding in overall hoarding behavior.

LL 40-42 “Hoarding behaviours are characterised by: the acquisition of, and failure 40 to discard, a large number of items that are of limited value; significant clutter in living spaces that render the activities associated with those spaces very difficult; and significant distress or impairment in functioning caused by the hoarding behaviours”

The clear distinction between “hoarding behaviour” and “hoarding disorder” is to discuss

LL 43 and further. The term “hoarding” is to avoid. Instead, use either “hoarding behaviour” or “hoarding disorder”, depending on what you mean.

LL 48-52 “Currently there is little information about people who hoard from normative community samples, as such individuals rarely come to the attention of research teams but estimates of its prevalence range from 1.5-6% [5] with a recent systematic review concluding that approximately 2 in every 100 people in the general population meet the criteria for hoarding disorder [6]”

For the prevalence of hoarding behaviour see Chaplin (4%). For the prevalence of hoarding behaviour AND hoarding disorder, see Nordsletten

LL 53 “Hoarding is a social [7], economic [8-9] and public health problem [10-11] and people who hoard experience a significant reduction in quality of life [12]”

First, the references are too old and contain general information on the burden of hoarding. See further works on the hoarding behaviour and risks of fire hazards, evictions, family relationships etc.

LL 54-56 Complaints of hoarding are addressed by multiple community services who have their own procedures and practices in relation to hoarding. In supported housing for example, people who hoard create a series of challenges relating to health and safety, risk management, and safeguarding [13]

See publications on the work of multi-professional teams in Laurentides, Quebec, Montreal, Quebec, and further, as well as hoarding task forces across the USA to put your work in the context of existing initiatives.

LL 57-59 In order to aid the development of a possible intervention for hoarding behaviours we aimed to identify current best practice by investigating key stakeholders’ existing practice with regard to identification, assessment and intervention associated with people who hoard

See the survey of primary mental health services conducted in Quebec, Canada, to put your work in the context of existing initiatives. The "best practices" is a term to avoid.

LL 63-64. We believe that, in accordance with Normalisation Process Theory [14], new interventions have the best chance of succeeding if they are based on an awareness and active engagement with existing organisational culture and practices.

As you refer to the Normalisation Process Theory, it should be presented, at least in scratch.

Materials and Methods

LL 72 “stakeholders”

The term is to explain shortly here.

International readers might misunderstand it.

Table 1.

The absence of building security and fire protection professionals among stakeholders is surprising. The police services seem to be underrepresented as well. To discuss in limits.

Discussion

The discussion mentions only one study out of many, discussing multi-professional initiatives in hoarding disorder. Large Canadian initiatives, already institutionalized, are overlooked (only in Canada: Saint-Jerome, Montreal, Vancouver, Edmonton).

6. PLOS authors have the option to publish the peer review history of their article (what does this mean?). If published, this will include your full peer review and any attached files.

Reviewer #1: **Yes: **Yuliya Bodryzlova

---

## [Author Response · Author response to Decision Letter 0]

20 Sep 2022

Reviewer #1: “In an ideal world that would be a multiagency service because you need everybody’s expertise.” Best practice for managing hoarding disorder: A qualitative investigation of existing procedures and practices. The study presents the attitudes and practices of key intervenors in case management of hoarding behaviour. Authors point out that the management of hoarding behaviour in the community is difficult, that attitudes and practices of stakeholders are homogeneous, and that creation of the multiprofessional team might be difficult. It is a timely and needed study.

Thank you for recognising the importance of our study. We appreciate your constructive feedback and have amended our manuscript in light of these comments. We hope you find the manuscript further improved and therefore now suitable for publication. Below is a point by point response to your comments.

Its only limit is that it omits the wide international experience in creating multi-professional teams for hoarding disorder around the world. An extensive body of academic and gray literature exists on the topic. Integrating this literature into the introduction and discussion would greatly improve the value of the current work.

Thank you for this comment. We were not aware of the body of published and grey literature on multi-agency teams for hoarding disorder, particularly the various Canadian initiatives. Based on the information you provided and via forward and backward citation searching we feel that we have now identified this literature, and this has now been added to the manuscript strengthening both the introduction and discussion.

In addition, the reading of the paper by a clinical psychologist is desirable in those which concern the distinction between hoarding behaviour and hoarding disorder, as well as more ancient and more recent cognitive models of hoarding disorder.

One of the authors (RC) is a clinical psychologist and she has carefully checked the paper particularly in terms of terminology and cognitive models of hoarding disorder. The manuscript relates to hoarding disorder (rather than behaviour) therefore we have amended terminology accordingly throughout the manuscript. Models of hoarding disorder have also been added.

Title and abstract.

“In an ideal world that would be a multiagency service because you need everybody’s expertise.” Best practice for managing hoarding disorder: A qualitative investigation of existing procedures and practices. Short Title: Best practice for managing hoarding disorder – The title is incorrect. The term “best” practice is the. I would suggest avoiding the term “best practice” as misleading: it evokes clinical guidelines and systematic reviews.

We have removed all reference to “best practice” (in both title, short title and throughout the manuscript) as suggested so as not to mislead the reader.

Introduction.

LL 35-36. “’ Possessions are accumulated over time and many of these items are given sentimental value”. See reasons for hoarding in Moulding, Knight, O'Connor. Also, see the cognitive model of hoarding in Frost and Hartl.

Reasons for accumulating possessions have been clarified and referenced along with suggested reasons for hoarding disorder with references (Moulding R, Kings C, Knight T. The things that make us: self and object attachment in hoarding and compulsive buying-shopping disorder, Current Opinion in Psychology. 2021; 39: 100-104/Frost RO, Hartl TL. A cognitive-behavioral model of compulsive hoarding. Behaviour Research and Therapy. 1996; 34 (4): 341-350)

LL 36-37. “such behaviour may be adaptive by ensuring survival when resources become scare”. Interesting tackle, but still unapproved. Sources are needed

The source of this theory has been added. (Grisham JR, Barlow DH. Compulsive Hoarding: Current Research and Theory. Journal of Psychopathology and Behavioral Assessment. 2005; 27(1): 45-52)

LL 38-39 However, in a minority of cases the normal hoarding tendency becomes pathological, and the person hoards uncontrollably. See Nordsletten for the exact proportion of problematic hoarding in overall hoarding behavior.

We have added and referenced the prevalence of hoarding disorder within this sentence using both the Nordsletten citation as suggested (Nordsletten A, Reichenberg A, Hatch S, De la Cruz L, Pertusa A, Hotopf M, Mataix-Cols D. Epidemiology of hoarding disorder. British Journal of Psychiatry. 2013; 203(6): 445-452.) and a more recent systematic review (Postlethwaite A, Kellett S, Mataix-Cols D. Prevalence of hoarding disorder: A systematic review and meta-analysis. Journal of Affective Disorders. 2019; 256: 309-316.)

LL 40-42 “Hoarding behaviours are characterised by: the acquisition of, and failure 40 to discard, a large number of items that are of limited value; significant clutter in living spaces that render the activities associated with those spaces very difficult; and significant distress or impairment in functioning caused by the hoarding behaviours”. The clear distinction between “hoarding behaviour” and “hoarding disorder” is to discuss

As previously stated, the manuscript relates to hoarding disorder (rather than behaviour) therefore we have amended terminology accordingly throughout the manuscript.

LL 43 and further. The term “hoarding” is to avoid. Instead, use either “hoarding behaviour” or “hoarding disorder”, depending on what you mean.

Thank you for identifying these inconsistencies in the terminology we have used in the manuscript. As previously stated, this manuscript relates to hoarding disorder, and we have amended the manuscript where possible to avoid use of the term “hoarding”. Please note that we have not made any amendments to participants direct quotations as these have been transcribed verbatim.

LL 48-52 “Currently there is little information about people who hoard from normative community samples, as such individuals rarely come to the attention of research teams but estimates of its prevalence range from 1.5-6% [5] with a recent systematic review concluding that approximately 2 in every 100 people in the general population meet the criteria for hoarding disorder [6]”. For the prevalence of hoarding behaviour see Chaplin (4%). For the prevalence of hoarding behaviour AND hoarding disorder, see Nordsletten

As previously stated, this manuscript relates to hoarding disorder. We have added and referenced the prevalence of hoarding disorder within the sentence on line 38-39 as suggested using the Nordsletten citation (Nordsletten A, Reichenberg A, Hatch S, De la Cruz L, Pertusa A, Hotopf M, Mataix-Cols D. Epidemiology of hoarding disorder. British Journal of Psychiatry. 2013; 203(6): 445-452.) and a more recent systematic review (Postlethwaite A, Kellett S, Mataix-Cols D. Prevalence of hoarding disorder: A systematic review and meta-analysis. Journal of Affective Disorders. 2019; 256: 309-316.). This sentence (line 48-52) has therefore been deleted.

LL 53 “Hoarding is a social [7], economic [8-9] and public health problem [10-11] and people who hoard experience a significant reduction in quality of life [12]” First, the references are too old and contain general information on the burden of hoarding. See further works on the hoarding behaviour and risks of fire hazards, evictions, family relationships etc.

Thank you for this suggestion. We have used more recent citations to support this sentence where necessary and have added some more specific consequences of hoarding disorder.

LL 54-56 Complaints of hoarding are addressed by multiple community services who have their own procedures and practices in relation to hoarding. In supported housing for example, people who hoard create a series of challenges relating to health and safety, risk management, and safeguarding [13] See publications on the work of multi-professional teams in Laurentides, Quebec, Montreal, Quebec, and further, as well as hoarding task forces across the USA to put your work in the context of existing initiatives.

Thank you for this comment. We were not aware of the body of published and grey literature on multi-agency teams for hoarding disorder, particularly the various Canadian initiatives. Based on the information you provided and via forward and backward citation searching we feel that we have now identified this literature, and this has now been added to the manuscript in both the introduction and discussion in order to put our work in the context of existing initiatives.

LL 57-59 In order to aid the development of a possible intervention for hoarding behaviours we aimed to identify current best practice by investigating key stakeholders’ existing practice with regard to identification, assessment and intervention associated with people who hoard. See the survey of primary mental health services conducted in Quebec, Canada, to put your work in the context of existing initiatives. The "best practices" is a term to avoid.

We have removed all references to “best practice” (in both title, short title and throughout the manuscript) as suggested so as not to mislead the reader. We have also added additional references, particularly in the discussion, to place our work in the context of existing initiatives including the survey of primary mental health services in Quebec, Canada.

LL 63-64. We believe that, in accordance with Normalisation Process Theory [14], new interventions have the best chance of succeeding if they are based on an awareness and active engagement with existing organisational culture and practices. As you refer to the Normalisation Process Theory, it should be presented, at least in scratch.

Thank you for this suggestion. A brief, basic description of Normalisation Process Theory has been added to the manuscript.

Materials and Methods

LL 72 “stakeholders”. The term is to explain shortly here. International readers might misunderstand it.

Apologies for this oversight, we have added a brief explanation of the term stakeholders.

Table 1. The absence of building security and fire protection professionals among stakeholders is surprising. The police services seem to be underrepresented as well. To discuss in limits.

Both housing and fire service professionals were well represented on our focus groups, but we agree that the police services were underrepresented however this has already been reported as a limitation.

Discussion

The discussion mentions only one study out of many, discussing multi-professional initiatives in hoarding disorder. Large Canadian initiatives, already institutionalized, are overlooked (only in Canada: Saint-Jerome, Montreal, Vancouver, Edmonton).

Thank you for this comment. As mentioned previously we were not aware of the body of published and grey literature on multi-agency teams for hoarding disorder, particularly the various Canadian initiatives. Based on the information you provided and via forward and backward citation searching we feel that we have now identified this literature, and this has now been added to the manuscript in both the introduction and discussion in order to put our work in the context of existing initiatives.

---

## [Decision Letter · Decision Letter 1]

21 Nov 2022

PONE-D-22-14253R1“In an ideal world that would be a multiagency service because you need everybody’s expertise.” Managing hoarding disorder: A qualitative investigation of existing procedures and practices.PLOS ONE

Dear Dr. Haighton,

Thank you for submitting your manuscript to PLOS ONE. After careful consideration, we feel that it has merit but does not fully meet PLOS ONE’s publication criteria as it currently stands. Therefore, we invite you to submit a revised version of the manuscript that addresses the points raised during the review process.

Your manuscript has been reassessed by the reviewer from the previous round, as well as one additional expert. As you will see from the comments, the reviewers acknowledge that the manuscript has improved significantly, but there remain some concerns which should be addressed before your manuscript is suitable for publication.

We look forward to receiving your revised manuscript.

Kind regards,

Dr Joseph Donlan

Senior Editor

PLOS ONE

Reviewers' comments:

Reviewer's Responses to Questions

**Comments to the Author**

1. If the authors have adequately addressed your comments raised in a previous round of review and you feel that this manuscript is now acceptable for publication, you may indicate that here to bypass the “Comments to the Author” section, enter your conflict of interest statement in the “Confidential to Editor” section, and submit your "Accept" recommendation.

Reviewer #1: (No Response)

Reviewer #2: (No Response)

2. Is the manuscript technically sound, and do the data support the conclusions?

Reviewer #1: Yes

Reviewer #2: Partly

3. Has the statistical analysis been performed appropriately and rigorously? 

Reviewer #1: N/A

Reviewer #2: N/A

4. Have the authors made all data underlying the findings in their manuscript fully available?

Reviewer #1: Yes

Reviewer #2: No

5. Is the manuscript presented in an intelligible fashion and written in standard English?

Reviewer #1: Yes

Reviewer #2: Yes

6. Review Comments to the Author

Reviewer #1: The work is much improved. There are some minor corrections left. Thank you very much for the efforts and time you invested in it.

*HD - hoarding disorder.

L21-23. There is an APA definition of HD; there is a discussion on the cut-off point dividing the clinical and non-clinical populations.

L35 – The term "feasibility" is to avoid, it has its reserved maining in health studies.

L 38-48. Rewrite. The idea is good; the text is difficult to understand.

L 49: “limited value” is to redefine. At least, you may put the word in the quotes. The processions are valuable for people with HD.

L 51 The distress is caused by the need to get rid of processions. Possessions themselves are the source of emotional security.

L52-58 I see no need for this phrase. It destroys the logic of your narration.

L70-73: the word "community" is used two times in the same phrase.

L 83-85. The paragraph on the NTP is not clear enough. Some further explications are needed.

L 96 professional interest?

Results

Results: prevalence term has its definition and is to avoid in your context. It would be better to use "number of cases in professional practice", or "number of referrals".

In your results, you are working with the part of the reality presented in your introduction. As such, high attention to HD caused by TV shows should be mentioned in the introduction first. For references, see Tolin and Frost.

Discussion

I would avoid the "definition of HD" as we already have one in the DSM-V. It may be said as "what is the level of the HD severity demanding intervention from …. (name of services).”

Ottawa's report "No room to spare" may give insight on the "ideal" organization of care for HD, as far as we are talking about the ideal world.

Reviewer #2: Thank you for the invitation to review this revised manuscript on community-based interventions for hoarding. The manuscript presents a qualitative study of two focus groups comprised of 17 professionals from mental health professions, housing, fire prevention, law, and protective services. Understanding more about community-based interventions for hoarding is of great societal importance.

The relevance/centrality of normalization process theory to this study was unclear. The definition given on p. 5 sounds good, but how the theory informed the data collection, analysis, or interpretation was not apparent. Lines 83-84 seem to suggest the importance of pilot-testing an intervention after articulating the specific points of intervention – those steps seem to have already been taken by some of the community partnerships, task forces, and response models cited in lines 63-65.

I did not review the original submission, but I can see that many citations have been added to reflect some of the published and grey literature on multidisciplinary intervention teams in other countries. The discussion of those other initiatives and research programs is helpful, but it also adds confusion about the value of the present study. Overall, the rationale and conceptual foundation for the study is unclear. The manuscript points to numerous community-based models for intervention in hoarding that are already being used in other countries. How does this study represent the next step in knowledge about this topic? Why is this type of focus group, with these stakeholders, using the normalization process theory the best path forward to stimulate the creation of the kind of intervention already being used in other countries?

The manuscript does not present much evidence that it relates to hoarding disorder rather than behaviour. Most community-based interventions address hoarding behaviour, as they typically do not assess the psychological factors – such as reasons for saving stuff, whether the stuff was saved intentionally or passively, etc. – that are required to make a diagnosis of hoarding disorder. Quite possibly, the mental health participants in these focus groups would be making hoarding disorder diagnoses, but fire prevention officers and housing officials typically do not conduct assessments that would lead to a diagnosis – such as the reason for accumulating possessions. Is More typically, they’re assessing conditions of the home related to health and safety and adequate maintenance. Without an assessment of the person living in the home (not just conditions of the home), it is not possible to make a diagnosis of any disorder. This was perhaps most evident in the emphasis on the Clutter Scale as a key assessment tool; the Clutter Scale assesses only clutter volume, not any of the reasons how the home came to be the way it is.

The revised manuscript is much improved in providing scholarly and grey literature citations, but the references do not always support the statements in the text. Sometimes, the text implies an empirical study, whereas the citation is for a review paper or chapter or theoretical paper. For example, line 59 implies that the Bratiotis & Woody paper establishes a heightened risk of squalor in hoarding cases, but it does not. Luu et al. (2018; doi: 10.1016/j.jocrd.2018.08.005), Dong et al. (2012; doi: 10.1177/0898264311425597) and John Snowdon’s work from Australia do establish that elevated risk.

Some of the basic information presented about hoarding is not correct. Line 38 implies that hoarding is a synonym for collecting, but it is quite distinct (see Nordsletten et al. 2013, doi: 10.1016/j.comppsych.2012.07.063). The diagnosis in DSM-5 is called hoarding disorder; compulsive hoarding syndrome was an earlier term that is no longer being used. More broadly, I was confused about the heading, “There is no consensus definition of hoarding disorder.” I think many would argue that DSM-5 presents that definition. Based on the quotations provided, it seems like what the stakeholders were discussing is how to use the word “hoarding” - and I would agree there is no consensus definition for what that means in community settings. What is the threshold for referring to conditions in a home as “hoarding”? The quotes seem to suggest a lack of consensus about that.

I also think it is misleading to suggest that publicity about hoarding, including media reports, increases the prevalence of hoarding disorder. Publicity may result in increased case finding or higher caseloads for stakeholders participating in this study, but it is hard to see how it would result in increased prevalence.

I wondered about the generalizability of the messages in this study. The manuscript states that this is the first study of community-based hoarding intervention practices in the UK. Why is that important? How might the UK’s context differ in relevant ways from the context in other countries (e.g., Canada, US, Australia) where this type of intervention has been going on for awhile? Similarly, the rationale for the study provided in lines 68-70 suggests the importance of translating the work done in other countries to the UK situation but does not articulate how this translation process might be of broader relevance beyond the UK. This generalizability seems important for a journal with an international readership.

The Discussion section would be strengthened by being more focused. It presents several interesting points about community-based interventions, but there is no sense of how each one contributes to a larger message.

A more minor question is how was “key” stakeholder defined? What steps were taken to ensure that all the key stakeholder categories relevant to hoarding were represented in the final sample?

7. PLOS authors have the option to publish the peer review history of their article (what does this mean?). If published, this will include your full peer review and any attached files.

Reviewer #1: **Yes: **Yuliya Bodryzlova

Reviewer #2: **Yes: **Sheila Woody

---

## [Author Response · Author response to Decision Letter 1]

14 Dec 2022

Reviewer #1: The work is much improved. There are some minor corrections left. Thank you very much for the efforts and time you invested in it. *HD - hoarding disorder.

We are pleased that you think the manuscript is improved and thank you for suggesting the further minor corrections which we have made. We hope you find the manuscript now suitable for publication.

L21-23. There is an APA definition of HD; there is a discussion on the cut-off point dividing the clinical and non-clinical populations.

Thank you, we have now changed the terminology in relation to our results L21-23 from “definition” to “understanding” and “prevalence” to “number of cases” so that there is no confusion with the APA definition of hoarding disorder and clinical prevalence.

L35 – The term "feasibility" is to avoid, it has its reserved maining in health studies.

Thank you for this comment. We understand your concern and have therefore removed any reference to the term “feasibility”.

L 38-48. Rewrite. The idea is good; the text is difficult to understand.

We have rewritten this paragraph and hope that it is now easier to understand.

L 49: “limited value” is to redefine. At least, you may put the word in the quotes. The processions are valuable for people with HD.

You are correct in highlighting that the possessions are considered valuable to people with hoarding disorder. Therefore, we have amended the phrase “of limited value” to “regardless of their actual value” in line with the DSM definition of hoarding disorder. This phrase has been amended in both the abstract and introduction.

L 51 The distress is caused by the need to get rid of processions. Possessions themselves are the source of emotional security.

Again, you are correct in highlighting that distress is caused by the need to get rid of possessions. Therefore, we have added that there is “a perceived need to save the items and distress associated with discarding them” in line with the DSM definition of hoarding disorder. This phrase has been amended in both the abstract and introduction. We have however retained the phrase “causing significant distress or impairment in functioning” as this is also in line with the DSM definition which states “hoarding causes clinically significant distress or impairment in social, occupational, or other important areas of functioning”.

L52-58 I see no need for this phrase. It destroys the logic of your narration.

Thank you for this suggestion. We have deleted this sentence.

L70-73: the word "community" is used two times in the same phrase.

Thank you for identifying this repetition. We have deleted “in the community at the end of this sentence”.

L 83-85. The paragraph on the NTP is not clear enough. Some further explications are needed.

Apologies if this paragraph was not clear, we have added some further explanation of NPT.

L 96 professional interest?

This is helpful, thank you, we have added that key stakeholders are people with a professional interest or concern in hoarding disorder.

Results

Results: prevalence term has its definition and is to avoid in your context. It would be better to use "number of cases in professional practice", or "number of referrals".

Thank you for this suggestion, we have changed the term “prevalence” to “number of cases”.

In your results, you are working with the part of the reality presented in your introduction. As such, high attention to HD caused by TV shows should be mentioned in the introduction first. For references, see Tolin and Frost.

Thank you for this comment, we have mentioned in the introduction the increased public awareness of hoarding disorder because of TV shows with appropriate references.

Discussion

I would avoid the "definition of HD" as we already have one in the DSM-V. It may be said as "what is the level of the HD severity demanding intervention from …. (name of services).”

Thank you, we have now changed the terminology used throughout the manuscript from “definition” to “understanding” and “prevalence” to “number of cases” so that there is no confusion with the APA/DSM-V definition of hoarding disorder and clinical prevalence.

Ottawa's report "No room to spare" may give insight on the "ideal" organization of care for HD, as far as we are talking about the ideal world.

Thank you for this suggestion, we have added some details from the “no room to spare” and another similar report in the conclusion section giving insight into the “ideal” organisation of care for hoarding disorder.

Reviewer #2: Thank you for the invitation to review this revised manuscript on community-based interventions for hoarding. The manuscript presents a qualitative study of two focus groups comprised of 17 professionals from mental health professions, housing, fire prevention, law, and protective services. Understanding more about community-based interventions for hoarding is of great societal importance. The relevance/centrality of normalization process theory to this study was unclear. The definition given on p. 5 sounds good, but how the theory informed the data collection, analysis, or interpretation was not apparent.

Thank you for this comment. NPT states that new interventions have the best chance of succeeding if they are based on an awareness and active engagement with existing organisational culture and practices therefore NPT influenced data collection methods, topic guide and data analysis. This has been added to the text.

Lines 83-84 seem to suggest the importance of pilot-testing an intervention after articulating the specific points of intervention – those steps seem to have already been taken by some of the community partnerships, task forces, and response models cited in lines 63-65.

The complex intervention that has been developed from this study is a psychology led multiagency model which, although having some similarities to other community partnerships, community response models, community task forces and collaborative agreements, is novel in that it is psychology led and therefore would need pilot testing. The novel nature of our complex intervention has been highlighted in the conclusion section. In addition, community partnerships, community response models, community task forces and collaborative agreements have not been translated into either UK policy or practice and therefore would need to be piloted in this context (with a very different health and social care system). We have clarified this further in the introduction.

I did not review the original submission, but I can see that many citations have been added to reflect some of the published and grey literature on multidisciplinary intervention teams in other countries. The discussion of those other initiatives and research programs is helpful, but it also adds confusion about the value of the present study. Overall, the rationale and conceptual foundation for the study is unclear. The manuscript points to numerous community-based models for intervention in hoarding that are already being used in other countries. How does this study represent the next step in knowledge about this topic?

Thank you for this comment. As outlined above the complex intervention that has been developed from this study is a psychology led multiagency model which although having some similarities to other community partnerships, community response models, community task forces and collaborative agreements, is novel in that it is psychology led. The novel nature of our complex intervention has been highlighted in the conclusion section. In addition, community partnerships, community response models, community task forces and collaborative agreements have not been translated into either UK policy or practice and therefore would need to be investigated in this context which has a very different health and social care system to the USA, Canada and Singapore. We have clarified this further in the introduction. In addition, many of the current models are buried within the grey literature and have not been formally evaluated (this has been highlighted in the text) so our study adds to the emerging body of evidence.

Why is this type of focus group, with these stakeholders, using the normalization process theory the best path forward to stimulate the creation of the kind of intervention already being used in other countries?

As stated above and within the manuscript this kind of intervention, while being used in other countries in various forms, has not been translated into either UK policy or practice. In addition, the UK has a very different health and social care system to the USA, Canada and Singapore where most of the other interventions have been developed. Focus groups were chosen as they provided socially negotiated practice examples and key holders were people with a professional interest or concern in hoarding disorder who were involved in some capacity with people with hoarding disorder. NPT influenced these data collection methods, topic guide and data analysis as NPT states that new interventions have the best chance of succeeding if they are based on an awareness and active engagement with existing organisational culture and practices. This has been clarified throughout the manuscript.

The manuscript does not present much evidence that it relates to hoarding disorder rather than behaviour. Most community-based interventions address hoarding behaviour, as they typically do not assess the psychological factors – such as reasons for saving stuff, whether the stuff was saved intentionally or passively, etc. – that are required to make a diagnosis of hoarding disorder. Quite possibly, the mental health participants in these focus groups would be making hoarding disorder diagnoses, but fire prevention officers and housing officials typically do not conduct assessments that would lead to a diagnosis – such as the reason for accumulating possessions. Is More typically, they’re assessing conditions of the home related to health and safety and adequate maintenance. Without an assessment of the person living in the home (not just conditions of the home), it is not possible to make a diagnosis of any disorder. This was perhaps most evident in the emphasis on the Clutter Scale as a key assessment tool; the Clutter Scale assesses only clutter volume, not any of the reasons how the home came to be the way it is.

Thank you for this comment. The manuscript focusses on hoarding disorder as the purpose of the psychology led multiagency model would be to allow for diagnosis of hoarding disorder as well as coordination of care based on each patient’s mental stability to cope with intervention. However, as you have rightly identified some of the agencies involved will be dealing with hoarding behaviour prior to any diagnosis however the consensus was that key stakeholders would benefit from a psychology led multiagency model to allow for diagnosis.

The revised manuscript is much improved in providing scholarly and grey literature citations, but the references do not always support the statements in the text. Sometimes, the text implies an empirical study, whereas the citation is for a review paper or chapter or theoretical paper. For example, line 59 implies that the Bratiotis & Woody paper establishes a heightened risk of squalor in hoarding cases, but it does not. Luu et al. (2018; doi: 10.1016/j.jocrd.2018.08.005), Dong et al. (2012; doi: 10.1177/0898264311425597) and John Snowdon’s work from Australia do establish that elevated risk.

Thank you for highlighting this. The citation supporting the statement on line 59 has now been changed to Luu M, Lauster N, Bratiotis C, Edsell-Vetter J, Woody SR. Squalor in community-referred hoarded homes. Journal of obsessive-compulsive and related disorders. 2018; 19: 66-71. All other references have been checked for appropriateness.

Some of the basic information presented about hoarding is not correct. Line 38 implies that hoarding is a synonym for collecting, but it is quite distinct (see Nordsletten et al. 2013, doi: 10.1016/j.comppsych.2012.07.063).

Thank you for highlighting this. We did not intend to imply that collecting and hoarding were the same thing. We have deleted the reference to collecting on line 38.

The diagnosis in DSM-5 is called hoarding disorder; compulsive hoarding syndrome was an earlier term that is no longer being used. More broadly, I was confused about the heading, “There is no consensus definition of hoarding disorder.” I think many would argue that DSM-5 presents that definition. Based on the quotations provided, it seems like what the stakeholders were discussing is how to use the word “hoarding” - and I would agree there is no consensus definition for what that means in community settings. What is the threshold for referring to conditions in a home as “hoarding”? The quotes seem to suggest a lack of consensus about that.

Thank you, we have deleted the sentence which refers to compulsive hoarding syndrome. We have also changed the terminology used throughout the manuscript from “definition” to “understanding” and “prevalence” to “number of cases” so that there is no confusion with the APA/DSM-V definition of hoarding disorder and clinical prevalence.

I also think it is misleading to suggest that publicity about hoarding, including media reports, increases the prevalence of hoarding disorder. Publicity may result in increased case finding or higher caseloads for stakeholders participating in this study, but it is hard to see how it would result in increased prevalence.

Thank you for this comment. We have added a sentence in to the introduction regarding how publicity about hoarding has increased the public recognition of hoarding disorder. We have also amended the results to read “increased identification of cases was reported to be because of publicity about the disorder and TV documentaries about hoarding disorder”.

I wondered about the generalizability of the messages in this study. The manuscript states that this is the first study of community-based hoarding intervention practices in the UK. Why is that important? How might the UK’s context differ in relevant ways from the context in other countries (e.g., Canada, US, Australia) where this type of intervention has been going on for awhile? Similarly, the rationale for the study provided in lines 68-70 suggests the importance of translating the work done in other countries to the UK situation but does not articulate how this translation process might be of broader relevance beyond the UK. This generalizability seems important for a journal with an international readership.

Thank you for highlighting this. We have added to the introduction and the discussion the importance of this research to the UK and how the UK’s context differs in relevant ways from the context in other countries (e.g. Canada, the USA, Australia) where this type of intervention has become institutionalised. We have also discussed the generalisability of our results.

The Discussion section would be strengthened by being more focused. It presents several interesting points about community-based interventions, but there is no sense of how each one contributes to a larger message.

Thank you for this suggestion. We have amended the discussion and conclusion sections in order to provide more focus and relevance to our findings. We hope you find these sections improved.

A more minor question is how was “key” stakeholder defined? What steps were taken to ensure that all the key stakeholder categories relevant to hoarding were represented in the final sample?

As stated in the manuscript, key stakeholders were defined as people with a professional interest or concern in hoarding disorder. Key stakeholders were identified from and via an existing hoarding research group (https://www.northumbria.ac.uk/about-us/academic-departments/psychology/research/health-and-wellbeing/hoarding-research/), a multidisciplinary group (48 members) which brings together academics from English Universities, stakeholders from the Local Authorities, Housing Associations, Charities, Social Care Services, Mental Health Services, the NHS, and Emergency Services. Many of the key stakeholders were already members of this group, however members were also called upon to identify further key stakeholders. While our sample of stakeholders was relatively small, there was a wide variety of organisations involved; however we have acknowledged as a limitation that there were some services which were not represented such as the ambulance service and the police.

---

## [Decision Letter · Decision Letter 2]

16 Jan 2023

PONE-D-22-14253R2“In an ideal world that would be a multiagency service because you need everybody’s expertise.” Managing hoarding disorder: A qualitative investigation of existing procedures and practices.PLOS ONE

Dear Dr. Haighton,

Thank you for submitting your manuscript to PLOS ONE. After careful consideration, we feel that it has merit but does not fully meet PLOS ONE’s publication criteria as it currently stands. Therefore, we invite you to submit a revised version of the manuscript that addresses the points raised during the review process.

We look forward to receiving your revised manuscript.

Kind regards,

Mohammed Ayalew

Academic Editor

PLOS ONE

Journal Requirements:

Reviewers' comments:

Reviewer's Responses to Questions

**Comments to the Author**

1. If the authors have adequately addressed your comments raised in a previous round of review and you feel that this manuscript is now acceptable for publication, you may indicate that here to bypass the “Comments to the Author” section, enter your conflict of interest statement in the “Confidential to Editor” section, and submit your "Accept" recommendation.

Reviewer #1: All comments have been addressed

Reviewer #2: (No Response)

2. Is the manuscript technically sound, and do the data support the conclusions?

Reviewer #1: Yes

Reviewer #2: Yes

3. Has the statistical analysis been performed appropriately and rigorously? 

Reviewer #1: N/A

Reviewer #2: N/A

4. Have the authors made all data underlying the findings in their manuscript fully available?

Reviewer #1: Yes

Reviewer #2: (No Response)

5. Is the manuscript presented in an intelligible fashion and written in standard English?

Reviewer #1: Yes

Reviewer #2: Yes

6. Review Comments to the Author

Reviewer #1: I have no comments. Thank you for addressing all comments and recommendations. Hope your results will contribute to the institutionalization of multidisciplinary teams for HD and the formal evaluation of their work.

Reviewer #2: Thank you for the invitation to review another version of this manuscript. It is much improved, particularly with regard to the rationale and conceptual underpinnings of the study. The explanation of NPT and how that relates to the study is much clearer now. The various miscommunications that I raised in my previous review have all been addressed. The Discussion section is now clear and succinct.

I have a few minor remarks that the authors may wish to address.

In the first few lines of the Introduction, the language makes it unclear just how the authors define “hoarding”. On line 39, the manuscript states, “humans show a strong tendency to hoard possessions”. The associated reference does not establish this statement as fact (and I suppose it depends on how one defines “hoard”). If this statement is broadly true, then why is the prevalence of hoarding so low? On line 41, the manuscript refers to “normal hoarding tendency”, although what this is remains unclear. I have seen hoarding defined rather broadly as keeping more of something than is needed in the moment (sorry I can’t find the reference). From this perspective, a home pantry or freezer represents hoarding. If this (or something like it) is the meaning of hoarding that the authors intend at the outset of the Introduction, it would be helpful to clarify that because most of the paper is focused on problematic accumulation of possessions.

Lines 71-72 state that “Many of these task forces have become institutionalised within [the US, Canada, and Singapore]”. I’m not sure what it means for a task force to “become institutionalized”, but I’m not sure that has happened in the US and Canada. (I can’t speak about the situation in Singapore.) There are numerous community responses to hoarding in various municipalities, but in practice these interventions have precarious funding and remain difficult to access. [As an aside, and as demonstration that the manuscript has ignited my interest and curiosity, I am aware of very few formal policies related to hoarding (the few examples that I can recall are limited to housing providers), although of course formal policies would be very helpful. Another difference between the situation in the US and Canada and what I think you’re proposing for the UK is that in the US and Canada, these community-based approaches are typically approached from the perspective of tenancy preservation or fire prevention rather than as health care.]

Line 259 makes reference to the Clutter Scale. I think this is referring to the Clutter Image Rating scale. If that’s correct, it would probably be better to use the full name so interested readers can more accurately access it online.

7. PLOS authors have the option to publish the peer review history of their article (what does this mean?). If published, this will include your full peer review and any attached files.

Reviewer #1: **Yes: **Yuliya Bodryzlova

Reviewer #2: **Yes: **Sheila R. Woody

---

## [Author Response · Author response to Decision Letter 2]

18 Jan 2023

Thank you for taking the time to look at our manuscript again, we have addressed the minor remarks and hope the manuscript is now suitable for publication.

Reviewer #1: I have no comments. Thank you for addressing all comments and recommendations. Hope your results will contribute to the institutionalization of multidisciplinary teams for HD and the formal evaluation of their work.

Thank you for all your help improving the manuscript.

Reviewer #2: Thank you for the invitation to review another version of this manuscript. It is much improved, particularly with regard to the rationale and conceptual underpinnings of the study. The explanation of NPT and how that relates to the study is much clearer now. The various miscommunications that I raised in my previous review have all been addressed. The Discussion section is now clear and succinct.

Thank you also for your help improving the manuscript.

I have a few minor remarks that the authors may wish to address.

In the first few lines of the Introduction, the language makes it unclear just how the authors define “hoarding”. On line 39, the manuscript states, “humans show a strong tendency to hoard possessions”. The associated reference does not establish this statement as fact (and I suppose it depends on how one defines “hoard”). If this statement is broadly true, then why is the prevalence of hoarding so low? On line 41, the manuscript refers to “normal hoarding tendency”, although what this is remains unclear. I have seen hoarding defined rather broadly as keeping more of something than is needed in the moment (sorry I can’t find the reference). From this perspective, a home pantry or freezer represents hoarding. If this (or something like it) is the meaning of hoarding that the authors intend at the outset of the Introduction, it would be helpful to clarify that because most of the paper is focused on problematic accumulation of possessions.

Thank you for these comments. The manuscript does indeed focus on hoarding disorder, but we wanted to start by explaining that the accumulation of possession can be a normal human response until it becomes problematic. To clarify this further we have removed the sentence on line 39 which stated that “humans show a strong tendency to hoard possessions” and have amended the wording of the next sentence so that it reads “accumulating possessions may be a learnt behaviour, ensuring survival when resources become scare”. In addition we have removed reference to “normal hoarding tendency” so this sentence now reads “However, in a minority of cases (1.5%-2% of the population) saving behaviour becomes excessive and disconnected from any apparent function or purpose, and the person hoards uncontrollably.”

Lines 71-72 state that “Many of these task forces have become institutionalised within [the US, Canada, and Singapore]”. I’m not sure what it means for a task force to “become institutionalized”, but I’m not sure that has happened in the US and Canada. (I can’t speak about the situation in Singapore.) There are numerous community responses to hoarding in various municipalities, but in practice these interventions have precarious funding and remain difficult to access. [As an aside, and as demonstration that the manuscript has ignited my interest and curiosity, I am aware of very few formal policies related to hoarding (the few examples that I can recall are limited to housing providers), although of course formal policies would be very helpful. Another difference between the situation in the US and Canada and what I think you’re proposing for the UK is that in the US and Canada, these community-based approaches are typically approached from the perspective of tenancy preservation or fire prevention rather than as health care.]

Thank you for this useful comment, we have removed the sentence suggesting community task forces have become institutionalised.

Line 259 makes reference to the Clutter Scale. I think this is referring to the Clutter Image Rating scale. If that’s correct, it would probably be better to use the full name so interested readers can more accurately access it online.

Thank you for this suggestion, we are indeed referring to the Clutter Image Rating Scale, so we have used the full name on line 259 as well as in the abstract.

---

## [Decision Letter · Decision Letter 3]

14 Feb 2023

“In an ideal world that would be a multiagency service because you need everybody’s expertise.” Managing hoarding disorder: A qualitative investigation of existing procedures and practices.

PONE-D-22-14253R3

Dear Dr. Haighton,

We’re pleased to inform you that your manuscript has been judged scientifically suitable for publication and will be formally accepted for publication once it meets all outstanding technical requirements.

Kind regards,

Mohammed Ayalew, MSc

Academic Editor

PLOS ONE

Additional Editor Comments (optional):

Reviewers' comments:

Reviewer's Responses to Questions

**Comments to the Author**

1. If the authors have adequately addressed your comments raised in a previous round of review and you feel that this manuscript is now acceptable for publication, you may indicate that here to bypass the “Comments to the Author” section, enter your conflict of interest statement in the “Confidential to Editor” section, and submit your "Accept" recommendation.

Reviewer #2: All comments have been addressed

2. Is the manuscript technically sound, and do the data support the conclusions?

Reviewer #2: (No Response)

3. Has the statistical analysis been performed appropriately and rigorously? 

Reviewer #2: (No Response)

4. Have the authors made all data underlying the findings in their manuscript fully available?

Reviewer #2: (No Response)

5. Is the manuscript presented in an intelligible fashion and written in standard English?

Reviewer #2: (No Response)

6. Review Comments to the Author

Reviewer #2: (No Response)

7. PLOS authors have the option to publish the peer review history of their article (what does this mean?). If published, this will include your full peer review and any attached files.

Reviewer #2: **Yes: **Sheila Woody

---

## [Editor Report · Acceptance letter]

28 Feb 2023

PONE-D-22-14253R3 

“In an ideal world that would be a multiagency service because you need everybody’s expertise.” Managing hoarding disorder: A qualitative investigation of existing procedures and practices. 

Dear Dr. Haighton:

I'm pleased to inform you that your manuscript has been deemed suitable for publication in PLOS ONE. Congratulations! Your manuscript is now with our production department. 

Kind regards, 

on behalf of

Mr Mohammed Ayalew 

Academic Editor

PLOS ONE